# Human Leukocyte Antigen-G is enriched in presence of trypanosome in the dermis of individuals exposed to *gambiense* Human African Trypanosomiasis in Guinea and Côte d'Ivoire

Alisé Lagrave[1☯], Aïssata Camara[2☯], Laure Gineau[1], Magali Tichit[3],
Firmin Bolivar Gnankou[4], Alseny M'mah Soumah[5], Mariame Camara[5], Martial N'Djetchi[6],
Justin Windingoudi Kaboré[5], Oumou Camara[5], Bamoro Coulibaly[4], Blé Sépé[4],
Valentin Nanan[4], Koffi Alain De Marie Kouadio[4], Louis N'Dri[4], Thomas Konan[6],
Jacqueline Milet[1], Salimatou Boiro[2], Christelle Travaillé[7], Aline Crouzols[7],
Nathalie Petiot[7], Hamidou Ilboudo[8], David Hardy[3], Ibrahim Sadissou[9],
Jean-Mathieu Bart[9], Mamadou Camara[5], Dramane Kaba[4], Mathurin Koffi[6],
Bruno Bucheton[9], Vincent Jamonneau[9], David Courtin[1,9☯], Brice Rotureau [2,7☯]*

**1** Institut de Recherche pour le Développement, UMR261 MERIT, Santé Internationale de la Mère et de l'Enfant, Paris, France, **2** Parasitology Unit, Institut Pasteur of Guinea, Conakry, Guinea, **3** Histopathology Platform, Institut Pasteur, Paris, France, **4** Unité de Recherche Trypanosomoses, Institut Pierre Richet, Bouaké, Côte d'Ivoire, **5** Programme National de Lutte contre la Trypanosomiase Humaine Africaine, Ministère de la Santé, Conakry, Guinea, **6** Laboratoire de Biodiversité et Gestion des Ecosystèmes Tropicaux, Université Jean Lorougnon Guédé, Daloa, Côte d'Ivoire, **7** Trypanosome Transmission Group, Trypanosome Cell Biology Unit, INSERM U1347, Department of Parasites and Insect Vectors, Institut Pasteur, Université Paris Cité, Paris, France, **8** Institut de Recherche en Sciences de la Santé - Unité de Recherche Clinique de Nanoro, Nanoro, Burkina-Faso, **9** Intertryp, Institut de Recherche pour le Développement - Centre de Coopération Internationale en Recherche Agronomique pour le Développement - University of Montpellier, Montpellier, France

☯ These authors contributed equally to this work.
* rotureau@pasteur.fr

## Abstract

Human leukocyte antigen-G (HLA-G) is an immunomodulatory molecule known to play a crucial role in immune tolerance and regulation. In the context of human African trypanosomiasis (HAT), higher soluble HLA-G levels were detected in the plasma of confirmed cases, representing a serological marker of *T. b. gambiense* infection. As trypanosomes also invade extravascular tissues, especially the skin, this study explored the potential role of HLA-G in the dermal immune response during *T. b. gambiense* infection. Blood and skin samples from 50 seronegative individuals, 45 seropositive suspects and 36 confirmed HAT cases, collected between 2018 and 2022 in endemic foci of Guinea and Côte d'Ivoire, were analyzed. Plasmatic and dermal levels of HLA-G proteins were quantified by ELISA and immuno-histochemistry, respectively, and compared to the trypanosome detection results in the same samples. The implication of soluble HLA-G plasma level as a biomarker of *T. b. gambiense* infection was confirmed. In the dermis, HLA-G isoforms were expressed either with a granular distribution or with in diffuse halos. Granular patterns of dermal

**Data availability statement:** All data are in the manuscript and/or supporting information files.

**Funding:** This work was supported by the Institut Pasteur (DH and BR) and the Institut Pasteur of Guinea (BR), the Institut de Recherche pour le Développement (DEPIST-THA project to DC), the French Government Investissement d'Avenir programme - Laboratoire d'Excellence "Integrative Biology of Emerging Infectious Diseases" (ANR-10-LABX-62-IBEID to BR), the French National Agency for Scientific Research (project ANR-18-CE15-0012 TrypaDerm to BR), and the Bill and Melinda Gates Foundation (www.gatesfoundation.org) through the Trypa-NO! Project (grant number INV-001785 to JMB, BB and VJ). The funders had no role in study design, data collection and analysis, decision to publish, or preparation of the manuscript.

**Competing interests:** The authors have declared that no competing interests exist.

HLA-G were directly associated with the presence of trypanosomes in the dermis. The presence of diffuse halos was correlated to higher sHLA-G levels in the plasma. In total, this study provides the first evidence of the involvement of HLA-G in the extravascular immune response against parasites, especially in the skin. It shows that HLA-G distribution in the extravascular compartment also represents a biomarker of trypanosome infection.

## Author summary

Human leukocyte antigen-G (HLA-G) is a molecule known to play a crucial role in immune regulation. In the context of sleeping sickness, higher soluble HLA-G levels were previously detected in the blood of confirmed cases. As trypanosomes also invade tissues, this study explored the potential role of HLA-G in the immune response against trypanosomes in the skin. Blood and skin samples from 50 negative controls, 45 suspects and 36 confirmed cases of sleeping sickness, collected between 2018 and 2022 in Guinea and Côte d'Ivoire, were analyzed. Blood and skin levels of HLA-G were quantified and compared to the trypanosome detection results in the same samples. The implication of soluble HLA-G level in blood as a biomarker of *T. b. gambiense* infection was confirmed. HLA-G isoforms were expressed either with a granular distribution or in diffuse halos in the dermis. Granules were directly associated with presence of trypanosomes in the dermis whereas diffuse halos were correlated to higher HLA-G levels in the plasma. In total, this study provides the first evidence of the involvement of HLA-G in the immune response against parasites in tissues, especially in the skin. It shows that HLA-G distribution in tissues also represents a biomarker of the infection.

## Introduction

Human African trypanosomiasis (HAT), also known as sleeping sickness, is a tsetse-borne parasitic neglected tropical disease [1]. *Trypanosoma brucei gambiense* is the agent of the slow-progressing form of the disease (gHAT) and threatens millions of people in 24 countries in western and central Africa [2]. Strengthened surveillance and control activities over the past twenty-five years succeeded in progressively reducing transmission, in view of the disease elimination targeted by 2030 by the WHO [2].

The typical clinical evolution of gHAT is characterized by two successive stages. The hemolymphatic stage (stage 1) is defined by dissemination of the parasites, initially via the lymphatics, to spread throughout the vascular system. The second stage (meningo-encephalitic stage) occurs when parasites invade the central nervous system causing motor and sensory dysfunction [3]. During stage 2, several interactions between the host and the parasite promote the permeability of the blood-brain barrier. This is associated with the permeation of cytokines, chemokines and

parasites into the central nervous system. Therefore, the activation of microglia due to the presence of parasitic proteins, the recruitment of leukocytes and the cytokine environment in the brain leads to an imbalance of the immune reaction that promotes neuropsychiatric disorders, body atrophy, coma and death without treatment [4–7]. It is important to stress that an infection caused by *T. b. gambiense* is chronic, with the first stage lasting for months or even years without any clinical symptoms and detectable parasitemia [8]. This can be followed by either a rapid progression of the disease to stage 2, or by a spontaneous recovery of affected individual [8–10].

Due to the limited sensitivity of current diagnostic tests, trypanosomes, which numbers fluctuate over the course of an infection, might not be detected, leading to an incorrect treatment decision [11]. This is particularly problematic in asymptomatic infections with parasite densities remaining below the diagnostic test detection threshold [12]. These undiagnosed individuals are exempt from any treatment and could constitute potential reservoirs of parasites for the tsetse [12]. This idea was reinforced by the observation of substantial number of extravascular trypanosomes in the dermis of unconfirmed seropositive subjects in Guinea [13–15]. Therefore, the immuno-genetic backgrounds of these individuals have been extensively explored in the recent years, in order to understand the underlying immune responses associated with the development of HAT, as well as to possibly identify biological markers for improving diagnosis [16–20].

Human Leukocyte Antigen-G (HLA-G) is a non-classical HLA class Ib antigen, with important immune-regulatory functions, that differs from classical HLA class I antigens by a low polymorphism, different splice variants and an expression restricted to only few tissues [21]. It is mainly expressed on extra-villous cytotrophoblasts in the placenta, where it mediates maternal-fetal immune tolerance during pregnancy. While expression of HLA-G is restricted in healthy tissues, pathological conditions can, however, induce HLA-G expression [22]. Several studies have shown that the induction of HLA-G expression during tumor development resulted in the occurrence of a worse disease outcome and/ or a disease spread [22]. The study of the role of HLA-G in the progression of different infections has also provided important insights [23–29]. Most of these studies were focused on describing the polymorphism of *HLA-G* alleles, particularly in the 3' untranslated region (3' UTR), which could affect its level of expression in the microenvironment and serve as a marker for disease predisposition. The role played by HLA-G in the variable clinical outcomes of gHAT has also been previously studied [29]. High plasma levels of soluble HLA-G molecules (sHLA-G) in unconfirmed gHAT seropositive suspects were especially correlated with the risk of developing the disease [30]. The present study aims to investigate the possible differences in HLA-G expression in the skin of HAT-seronegative and HAT-seropositive subjects in Guinea and Côte d'Ivoire, as compared to their sHLA-G levels.

## Methods

### Ethical approval

All investigations were conducted in accordance with the Declaration of Helsinki and fulfil the STROBE criteria. Approval for this study were obtained from the Comité National d'Ethique pour la Recherche en Santé of the Republic of Guinea (authorization Study Diag-Cut-THA 032/CNERS/17 and amendment 038/CNERS/19) and from the Comité National d'Ethique des Sciences de la Vie et de la Santé of the Republic of Côte d'Ivoire (authorization #111–19/MSHP/CNESV-kp and amendments). Children under 16 years of age and pregnant women were excluded from the study. Each participant was informed about the study's objectives in their own language and provided written informed consent. For participants between 16–18 years of age, written informed consent was also obtained from their parents. The data of individual study participants were randomly anonymized with a 4-digit code at enrolment by the MD in charge of the medical campaign.

### Study enrolment, screening, and case definitions

The study was conducted in two study sites in Guinea (Boffa and Forecariah) and two in Côte d'Ivoire (Bonon and Sinfra) between December 2018 and April 2022. As described in detail in [14] and [15] (S1 Table), all subjects were enrolled

during medical surveys performed by the HAT National Control Programme in Guinea and the HAT National Elimination Programme in Côte d'Ivoire, according to WHO recommendations.

All enrolled individuals were first tested with the card agglutination test for trypanosomiasis using whole blood samples (CATTwb), and/ or one rapid diagnostic test for HAT on blood (SD Bioline HAT, Abbott Bioline HAT 2.0 or HAT Sero-K-SeT). For those individuals who tested positive in the serological screening test, 5 ml of blood were collected in heparinized tubes, and a two-fold plasma dilution series was used to determine the CATT plasma (CATTp) end titre. All individuals with CATTp end titres of 1/4 or higher underwent a microscopic examination of lymph node aspirate, whenever cervical swollen lymph nodes were present. Blood samples of CATTp-positive individuals were then centrifuged to obtain the buffy coat layer, which was tested for the presence of trypanosomes using the mini-anion exchange centrifugation test (mAECT BC) [31]. If trypanosomes were detected using this test, the infected individual underwent a lumbar puncture and their disease stage was determined by searching for trypanosomes using the modified simple centrifugation technique for cerebrospinal fluid (CSF) and with white blood cell (WBC) counts [32]. *Gambiense* HAT patients were classified as stage 1 (0–5 WBC/ µl and absence of trypanosomes in CSF) or stage 2 (> 5 WBC/ µl and/ or presence of trypanosomes in CSF) and were treated accordingly by the National Control Programmes. For stage 1 patients, treatment consisted of Pentamidine (intramuscular injection of 4 mg/ kg once daily for 7 days in adults) or Fexinidazole (oral doses of 1,800 mg fexinidazole once per day on days 1–4, then 1,200 mg fexinidazole on days 5–10) [33] or Acoziborole (1 oral dose of 960 mg) [34]. For stage 2 patients, treatment consisted of Nifurtimox-Eflornithine Combination Therapy (NECT) (oral Nifurtimox at 15 mg/ kg per day in three doses for 10 days and intravenous Eflornithine (α-difluoromethylornithine or DFMO) at 400 mg/ kg per day in two 2 h infusions for 7 days in adults) or Fexinidazole (oral doses of 1,800 mg once per day on days 1–4 then 1,200 mg on days 5–10) [33] or Acoziborole (1 oral dose of 960 mg) [34]. All parasitologically confirmed cases were diagnosed and treated according to WHO recommendations at that time and to the availability of the different treatments.

In summary, subjects were included in either of these 3 groups according to their diagnostic results at enrolment only (S1 Table), and whatever their HAT medical history:

- Seronegative controls (CTR): negative in CATTwb and/ or RDT, and negative in CATTp. In this group, subjects from Guinea have no HAT medical history. In Côte d'Ivoire, this group also includes previous cases confirmed and treated before 2019, previous cases confirmed before 2019 but left untreated (refusal or lost-to-follow-up after diagnosis), individuals found seropositive in CATT and in TL at least once before 2019 but who remained negative in parasitology, and individuals found seropositive in CATT at least once before 2019 but negative in both TL and parasitology, as described in [15].

- Unconfirmed seropositive suspects (SERO): positive in CATTwb and/ or RDT, and positive in CATTp, but negative in parasitological examination.

- Confirmed stage 1 case (HAT S1): positive in CATTwb and/ or RDT, and positive in CATTp, and positive in blood parasitological examination, but negative in CSF examination.

- Confirmed stage 2 case (HAT S2): positive in CATTwb and/ or RDT, and positive in CATTp, and positive in blood parasitological examination, and positive in CSF.

## Field procedure and sampling

As in [13–15], at enrolment as well as at each subsequent follow-up visit, each participant underwent an epidemiological interview and a clinical examination during which dermatological symptoms were assessed by a trained dermatologist. Follow-up visits were not strictly planed at the beginning of the study but rather organized according to the activities of the National Programmes. Only the following epidemiological data were considered in the present study: age and sex. Dermatological signs of pruritus (skin itch) and dermatitis (skin inflammation) were also investigated, and a careful examination

of the entire body was performed, to detect any symptoms that might be related to skin infections. Finally, two superficial skin snip biopsies were sampled in sterile conditions from the right back shoulder of all enrolled subjects. Biopsies were performed under local anaesthesia and were rapidly dressed. Two skin snips were then rapidly fixed in 10% neutral buffered formalin for immuno-histochemistry (IHC). Plasma aliquots from blood samples were also obtained during the screening step for use in serological trypanolysis tests, as described below.

## Quantitative ELISA

Plasmatic sHLA-G levels were quantified by ELISA enzyme assay as previously described [27,35]. The capture antibody was the mouse monoclonal anti-HLA-G MEM-G/9 IgG1 (1/ 10,000, Exbio, 11–292-M001) which recognizes the sHLA-G1 and G5 isoforms. The detection antibody was the rabbit anti-human $\beta 2$-microglobulin (1/ 10,000, Dako, A0072). The supernatants of transfected M8-HLA-G5 and M8 cell lines (M8-pcDNA) constituted the standard and negative controls, respectively [36]. The different incubations were performed at room temperature (about 22°C), and the washing solution was composed of Phosphate Buffer Saline (PBS 1X) with 0.05% Tween.

Briefly, 96-well microplates were incubated with MEM-G9 (dilution 1/ 100) overnight at 4°C. Unbound antibodies were removed the next day after 4 washes followed by incubation with 300 µl/ well of Dako diluent (Dako) for 2 h to saturate the medium. Plates were then washed 4 times and incubated for 2 h with 507 ng/ mL standard reagent (serially diluted from 0.78 to 100 ng/ mL) and samples (dilution 1/ 4). After 4 washes, soluble HLA-G antigens were detected in the wells by the addition of $\beta$-microglobulin (dilution 1/ 10,000) and left to incubate for 1 h. Plates were washed again 4 times and incubated with Envision enzyme (Dako, dilution 1/ 200) for 1 h. A final wash of the plates followed by the addition of Tetra-Methyl-Benzidine substrate (Sigma Adrich) was performed. The plates were kept in the dark for 45 min, and the reaction was stopped by adding HCL. The absorbance was read at 450 nm with the Multiskan FC Photometer (Thermo Scientific Microplate reader). Total sHLA-G concentrations were calculated from the standard curve at 8 dilutions. Each sample was tested in duplicate with a 100 ng/ mL M8-HLA-G5 positive control and a M8-pcDNA negative control (dilution 1/ 5).

## Immune trypanolysis test

A plasma sample from all participants was used to perform the immune trypanolysis test. This test detects complement-mediated immune responses activated by either the LiTat 1.3, LiTat 1.5 or LiTat 1.6 variable surface antigens specific for *T. b. gambiense*, as previously described [37].

## Immunohistochemical detection (IHC) of HLA-G in the dermis

Skin snip biopsy samples fixed in 10% formalin and preserved at 4 °C were trimmed and processed into paraffin 60 blocks in the lab. Longitudinal sections of ~2.5 µm were prepared and processed using Dako Autostainer Plus (Dako).

To detect the presence of trypanosomes, sections were first immunolabelled with the *T. brucei*-specific anti-ISG65 antibody that targets the Invariant Surface Glycoprotein 65 (rabbit, 1/ 800 dilution; gift from M. Carrington, Cambridge, UK) [38] as described in [13–15].

To detect HLA-G molecules, a blocking step was first performed with a 3% BSA solution (Sigma). The primary antibody used was the mouse IgG anti-HLA-G 4H84 (1/ 400, Exbio, 11–499-C100). The 4H84 antibody specifically detects human HLA-G and recognizes an epitope within 23 amino acids from the alpha 1 region of all HLA-G isoforms. As a secondary antibody, the Polymer Expertise, Bond Polymer Refine Detection kit (Leica, DS9800) was used with a pH 6 citrate unmasking buffer (Leica, HIER 1_20mn). For each sample, a negative control was performed with PBS 1X and the Bond Polymer Leica Kit only (no primary antibody). Positive control staining was performed on human placenta sections collected during the STOPPAM project in the Mono province, Benin [39] (S1 Fig). Technical negative and positive controls were systematically done in parallel of each batch of slides processed together. Immunostaining images were

acquired using an automated Axio Observer Z1 microscope (Carl Zeiss) and analysed using the ZEN 3.9 software (Carl Zeiss).

Semi-quantitative assessment of HLA-G loads in skin sections was remotely assessed by two independent biologists blind to group assignment and experimental procedures. Different parameters were first quality-checked to validate the exploitability of each slide. The readability of the slide was assessed by the integrity of the overall structure of the section. Then, we checked that the image definition was compatible with HLA-G detection. Non-readable images (blurry, out of focus, over/ under exposed) were re-scanned once. Then, the presence of a minimal amount of dermis was also checked by controlling the thickness of the section (excluding the epidermis and an inward thickness equivalent to an epidermis to avoid any source of melanocytic contamination). Non-validated slides were not further considered for HLA-G detection. In quality-check validated slides, the presence of HLA-G was detected by observing granular brown spots and/ or diffuse halos at x 40 magnification (Fig 1). The numbers of granular brown spots and diffuse halos per slide were quantified using the ZEN 3.9 software (Carl Zeiss) and deltas were calculated between two successive anti-HLA-G labelled slide and unstained negative control slide for each individual skin sample. In total, a validated slide was considered positive for the presence of HLA-G when the delta of the sum of diffuse halos and/ or the delta of the sum of granular brown spots were equal or greater than 1. This allowed us to the assess the presence of soluble and/ or membrane HLA-G in the dermal region of most samples.

### Data analyses

Soluble HLA-G plasmatic concentration was used as a quantitative variable. At enrolment, we used linear regression model to compare log transformation of sHLA-G + 1 to linearity assumption with different explanatory variables with the VGAM package in R software version 4.4.1. For follow-up data, linear mixed effects model was used. HLA-G expression in diffuse halos or granular brown spots were used as categorical variables, dichotomic (presence versus absence) or multinomial (null [0], medium (]0,3] for diffuse halos and}0,2] for granular brown spots), and high expression (]3,168] for diffuse halos and}2,28] for granular brown spots)), based on the tertiles of the distribution. At enrolment, we used logistic regression and multinomial logistic regression to compare dichotomic and multinomial variables, respectively, with the lmer4 package in R software version 4.4.1. Other boolean variables (mAECT and TL results) were analyzed with logistic regression model with the lmer4 package in R software version 4.4.1. For follow-up data, a generalized linear mixed effects model was used to dichotomic variable with the lmer4 package in R software version 4.4.1. All variables were initially introduced into the multivariate model. All analyses were adjusted on age, sex, occurrence of fever, HAT transmission focus, occurrence of pruritus and occurrence of dermatitis. The final model was selected through a backward procedure and only covariates with P value < 0.5 were retained in the final model.

## Results

### Population and demographics

A total of 131 individuals were included in the study (42 in Boffa, 41 in Forecariah, 30 in Bonon and 18 in Sinfra), and a total of 218 blood and skin samples were collected at 1–3 timepoints per subject (78 in Boffa, 65 in Forecariah, 46 in Bonon and 29 in Sinfra). At enrolment, a total of 128 samples were successfully analyzed in ELISA for HLA-G, 113 samples in IHC for HLA-G, and 98 samples in IHC for dermal trypanosomes (anti-ISG65). A description of the study population is provided in Table 1 and S2 Fig. Most clinical symptoms observed in confirmed HAT cases were not specific and their main clinical features are detailed in [13–15]. The two most frequent symptoms were the occurrence of swollen cervical lymph nodes and dermatological signs (dermatitis and/ or pruritus). All data are provided in S1 File.

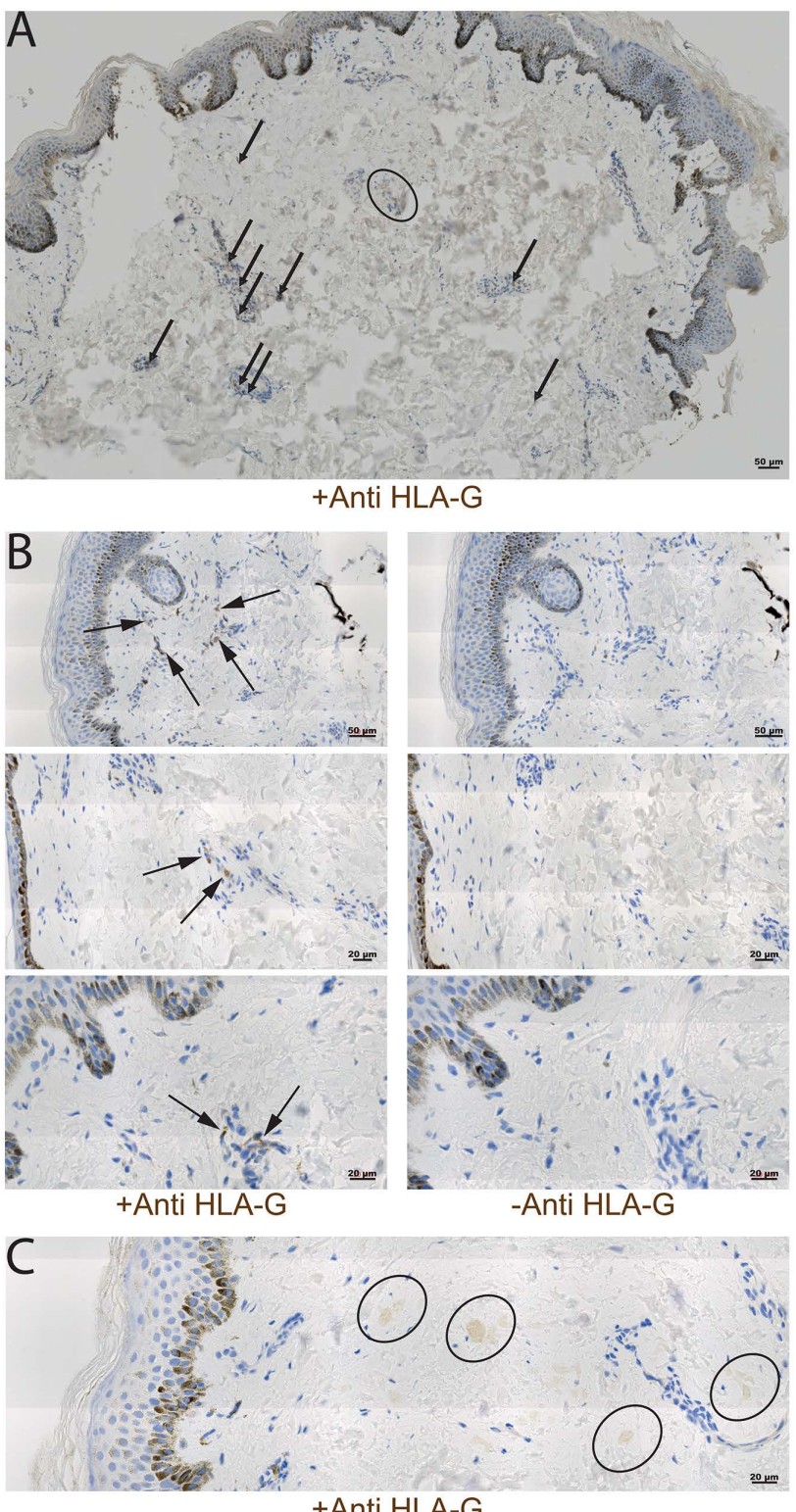

**Fig 1. Immuno-histochemical detection of HLA-G in the dermis.** Formalin-fixed paraffin-embedded skin sections were immunolabelled with the anti-HLA-G 4H84 antibody. This figure shows 5 distinct samples. **(A)** Image of an entire skin section from a confirmed HAT case showing the typical HLA-G localization as diffuse halos (circle) and granular brown spots (arrows). The darker band on the top of the section corresponds to the melanized

epidermis, and the white region to the dermis. **(B)** Higher magnification images showing dark granular HLA-G structures indicated with arrows in labelled skin sections (left) as compared to the corresponding unstained control slides (right). These three skin samples originate from confirmed HAT cases. **(C)** Skin section from a seropositive subject with HLA-G detected in diffuse halos (circle).

## Dermal trypanosomes

Results of the dermal trypanosome detection in IHC at enrolment are described in Table 1. At enrolment, after analysis of 98 validated slides in IHC, 29.6% of the skin samples were found positive for the presence of trypanosomes. When considering all timepoints, as previously observed, no significant association was found between the presence of parasites in blood and the presence of parasites in the dermis. Similarly, no significant association was found between results of the trypanolysis tests and the presence of parasites in the dermis (S2 Table).

## Plasmatic levels of sHLA-G are associated with individual HAT status

Plasmatic sHLA-G levels at enrolment are described in Table 1. Soluble HLA-G concentrations were quantified in the plasma samples collected in the three different general groups, the control (CTR) group (n = 50), the seropositive (SERO) group (n = 45) and the HAT cases (HAT) group (n = 36). Seronegative controls (CTR) included individuals who were all negative in CATTwb and/ or RDT, and all negative in CATTp at enrolment. Among subjects in the CTR group, preliminary univariate analyses showed no difference neither in sHLA-G levels nor in dermal HLA-G profiles between naïve subjects and subjects with a known history of HAT. The seropositive and HAT groups were respectively divided in two specific subgroups. Seropositive participants (CATT+ titer ≥ ¼) were subdivided according to the result of the immune trypanolysis test in CATT+ TL- (SERO/TL-) (n = 38) and CATT+ TL+ (SERO/TL+) (n = 7). In the HAT group, patients were either subdivided in stage 1 (S1) (n = 8) or in stage 2 (S2) (n = 28).

At enrolment, the results obtained by linear regression model showed significantly higher titers of plasmatic sHLA-G in the HAT (n = 36) ($p = 3.4*10^{-6}$) and SERO (n = 45) ($p = 3.78*10^{-3}$) groups than in the CTR group (n = 50) (Fig 2A and Table 2A). A higher sHLA-G plasmatic concentration was observed in the HAT group as compared to the SERO group ($p = 0.016$) (Table 2A). Compared to the CTR group, significantly higher sHLA-G levels were observed in the subgroups SERO/TL- (n = 38) ($p = 0.037$), SERO/TL+ (n = 7) ($p = 4.89*10^{-5}$), S1 (n = 8) ($p = 1.37*10^{-3}$) and S2 (n = 28) ($p = 3.83*10^{-6}$) (Fig 2B and Table 2B). Moreover, SERO/TL- subjects showed lower sHLA-G levels as compared to those in the SERO/TL+ ($p = 2.94*10^{-3}$), S1 ($p = 0.047$) and S2 ($p = 1.95*10^{-3}$) subgroups (Fig 2B and Table 2B). No significant difference of plasmatic sHLA-G concentration was observed between subjects in the SERO/TL+, S1, and S2 subgroups. No significant difference was neither observed between patients in the S1 and S2 subgroups (Fig 2B). In contrast, and as previously observed, high levels of sHLA-G were significantly associated with the presence of parasites in the blood detected in mAECT ($p = 1.18*10^{-5}$) (Section A in S3 Table), and with a positive result in immune trypanolysis test ($p = 9.86*10^{-5}$) (Section A in S4 Table).

Details of sHLA-G levels in samples collected during the follow-up are described in Table 3. The results obtained by linear regression mixed model showed significantly higher sHLA-G plasmatic concentrations among subjects in the SERO group (n = 83) ($p = 8.13*10^{-4}$) and in the HAT group (n = 38) ($p = 1.8*10^{-4}$) than in the CTR group (n = 83) (Fig 3A and Table 4A). However, no significant difference in plasma sHLA-G levels was observed between the SERO and HAT groups (Fig 3A and Table 4A). Compared to subjects in the CTR group, significantly higher sHLA-G levels were observed in the subgroups SERO TL- (n = 52) ($p = 0.033$), SERO TL+ (n = 24) ($p = 0,01$) and S2 (n = 30) ($p = 3.37*10^{-6}$) (Fig 3B and Table 4B). A suggestive, yet non-significant, difference was observed between the CTR group and the S1 subgroup (n = 8) ($p = 0.053$), with higher levels of sHLA-G in the S1 subgroup (Fig 3B and Table 4B). Significantly higher levels of plasmatic sHLA-G were observed in the S2 as compared to the SEROTL- subgroup ($p = 0.017$) (Fig 3B and Table 4B). In addition, high levels of sHLA-G were also

**Table 1. Characteristics of study participants, plasmatic and dermal HLA-G levels, and dermal trypanosomes detection at enrolment.**

| Phenotypic groups | | Sex | Age | Focus | Plasmatic sHLA-G | Dermal HLA-G diffuse halos in IHC | Dermal HLA-G granular brown spots in IHC | Presence of dermal trypanosomes in IHC |
|---|---|---|---|---|---|---|---|---|
| | | Number | Number | | Number | Number | Number | Number |
| | | % (No.) | Mean [Range], years | % (No.) | Mean [SEM], ng/ml | % (No.) | % (No.) | % (No.) |
| CTR | All | n = 50 | n = 50 | Boffa: 6.00 (3) | n = 48 | n = 37 | n = 37 | n = 32 |
| | | Female: 54.00 (27) | 41.30 [18-89] | Forecariah: 26.00 (13) | 6.78 [5.81] | Neg: 29.73 (11) | Neg: 51.35 (19) | Neg: 68.75 (22) |
| | | Male: 46.00 (23) | | Bonon: 40.00 (20) | | Pos: 70.27 (26) | Pos: 48.65 (18) | Pos: 31.25 (10) |
| | | | | Sinfra: 28.00 (14) | | | | |
| SERO | All | n = 45 | n = 45 | Boffa: 53.33 (24) | n = 44 | n = 44 | n = 44 | n = 39 |
| | | Female: 55.56 (25) | 40.00 [16-70] | Forecariah: 17.78 (8) | 10.14 [6.91] | Neg: 50.00 (22) | Neg: 56.81 (19) | Neg: 64.10 (25) |
| | | Male: 44.44 (20) | | Bonon: 22.22 (10) | | Pos: 50.00 (22) | Pos: 43.19 (25) | Pos: 35.90 (14) |
| | | | | Sinfra: 6.67 (3) | | | | |
| | SERO TL- | n = 38 | n = 38 | Boffa: 55.26 (21) | n = 37 | n = 37 | n = 37 | n = 33 |
| | | Female: 57.90 (22) | 40.21 [16-70] | Forecariah: 13.16 (5) | 8.42 [5.12] | Neg: 51.35 (19) | Neg: 43.24 (16) | Neg: 66.67 (22) |
| | | Male: 42.10 (16) | | Bonon: 23.68 (9) | | Pos: 48.65 (18) | Pos: 56.76 (21) | Pos: 33.33 (11) |
| | | | | Sinfra: 7.89 (3) | | | | |
| | SERO TL+ | n = 7 | n = 7 | Boffa: 42.86 (3) | n = 7 | n = 7 | n = 7 | n = 6 |
| | | Female: 42.86 (3) | 36.43 [20-60] | Forecariah: 42.86 (3) | 19.23 [8.37] | Neg: 42.86 (3) | Neg: 42.86 (3) | Neg: 50.00 (3) |
| | | Male: 57.14 (4) | | Bonon: 14.29 (1) | | Pos: 57.14 (4) | Pos: 57.14 (4) | Pos: 50.00 (3) |
| | | | | Sinfra: 0 (0) | | | | |
| HAT | All | n = 36 | n = 36 | Boffa: 41.67 (15) | n = 36 | n = 32 | n = 32 | n = 27 |
| | | Female: 55.56 (20) | 33.33 [12-60] | Forecariah: 55.55 (20) | 18.29 [12.65] | Neg: 50.00 (16) | Neg: 46.88 (15) | Neg: 81.48 (22) |
| | | Male: 44.44 (16) | | Bonon: 0 (0) | | Pos: 50.00 (16) | Pos: 53.12 (17) | Pos: 18.52 (5) |
| | | | | Sinfra: 2.78 (1) | | | | |
| | S1 | n = 8 | n = 8 | Boffa: 12.5 (1) | n = 8 | n = 8 | n = 8 | n = 8 |
| | | Female: 50.00 (4) | 39.12 [16-53] | Forecariah: 87.5 (7) | 19.44 [16.55] | Neg: 62.5 (5) | Neg: 75 (6) | Neg: 80.00 (6) |
| | | Male: 50.00 (4) | | Bonon: 0 (0) | | Pos: 37.5 (3) | Pos: 25 (2) | Pos: 20.00 (2) |

*(Continued)*

Table 1. (Continued)

| Phenotypic groups | Sex | Age | Focus | Plasmatic sHLA-G | Dermal HLA-G diffuse halos in IHC | Dermal HLA-G granular brown spots in IHC | Presence of dermal trypanosomes in IHC |
|---|---|---|---|---|---|---|---|
| | | Number | | Number | Number | Number | Number |
| | | % (No.) | Mean [Range], years | | % (No.) | Mean [SEM], ng/ml | % (No.) | % (No.) | % (No.) |
| | | | | Sinfra: 0 (0) | | | | |
| | S2 | n=28 | n=28 | Boffa: 50.00 (14) | n=28 | n=24 | n=24 | n=22 |
| | | Female: 57.14 (16) | 31.68 [12-60] | Forecariah: 46.43 (13) | 17.96 [11.66] | Neg: 45.83 (11) | Neg: 37.50 (9) | Neg: 81.82 (18) |
| | | Male: 42.86 (12) | | Bonon: 0 (0) | | Pos: 54.17 (13) | Pos: 62.50 (15) | Pos: 18.18 (4) |
| | | | | Sinfra: 3.57 (1) | | | | |
| Total | All | n=131 | n=131 | Boffa:32.06 (42) | n=128 | n=113 | n=113 | n=98 |
| | | Female: 54.96 (72) | 38.69 [12-89] | Forecariah: 31.30 (41) | 11.17 [9.74] | Neg: 43.36 (49) | Neg: 46.90 (53) | Neg: 70.41 (69) |
| | | Male: 45.04 (59) | | Bonon: 22.90 (30) | | Pos: 56.64 (64) | Pos: 53.10 (60) | Pos: 29.59 (29) |
| | | | | Sinfra: 13.74 (18) | | | | |

CTR, control; SERO, seropositive; TL, trypanolysis; HAT, human African trypanosomiasis confirmed cases; S1, stage 1; S2, stage 2; SEM, Standard error of the mean; sHLA-G, soluble human leukocyte antigen-G; Neg, negative; Pos, positive.

significantly associated with the presence of parasites in the blood detected in mAECT (p=2.86*10⁻³) (Section B in S3 Table), and with a positive result in immune trypanolysis test (p=2.25*10⁻³) (Section B in S4B Table).

### HLA-G protein expression in the dermis according to HAT status

When considering all timepoints, HLA-G molecules were detected in the dermis of up to 64 samples out of 113 tested (57% with diffuse halos, Tables 1 and 3 and Fig 1). At enrolment, the results obtained by logistic regression model showed no significant difference in the proportion of positive slides, neither for HLA-G delta granular brown spots, nor for HLA-G delta diffuse halos, between the general status groups and subgroups (Table 1). Interestingly, however, a higher proportion of skin samples positive for HLA-G granular brown spots was observed in presence of dermal parasites detected in IHC (anti-ISG65) at enrolment (p=3.16*10⁻³) (Table 5A), and during follow-up visits (p=2.5*10⁻³) (Table 5B). A similar significant association was also observed between the positivity for HLA-G granular brown spots in a dermal sample and the occurrence of dermatitis in the subject (Table 5). However, no significant association was observed between the proportion of slides positive for HLA-G delta diffuse halos and the presence of parasites in the dermis assessed by IHC.

### Correlation between HLA-G protein expressions in the plasma and in the dermis

At enrolment, the results obtained by linear regression models showed no association between plasmatic sHLA-G levels and the proportion of samples positive for HLA-G granular brown spots in the dermis. In contrast, high

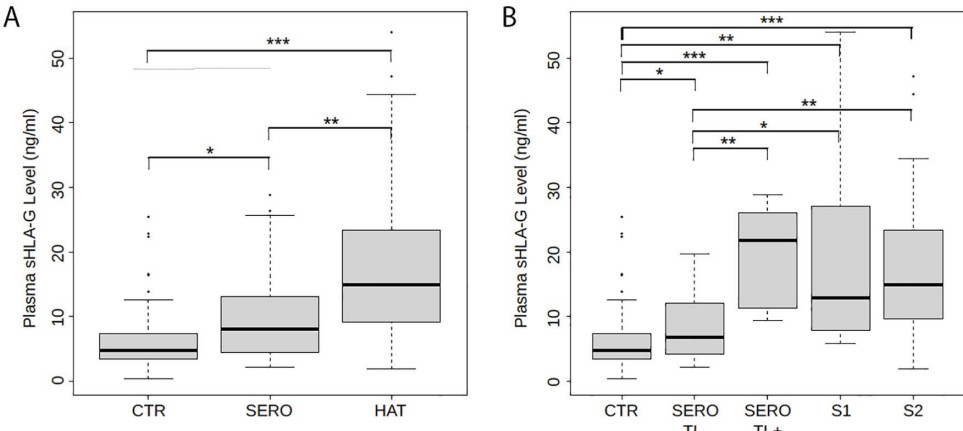

**Fig 2. Plasmatic sHLA-G concentrations by HAT status at enrolment. A)** Plasma levels in control (CTR), seropositive (SERO) and HAT groups. **B)** Plasma levels in CTR, SERO and negative trypanolysis test (SERO/TL-), SERO and positive trypanolysis test (SERO/TL+), stage 1 (S1) and stage 2 (S2) subgroups. *P <. 05, ** P <.01, *** P <.001 (P values based on multinomial linear regression for associations between explanatory variables sHLAG level and HAT status adjusted on sex, age, focus, fever, pruritus and dermatitis status).

**Table 2. Associations between sHLA-G plasmatic levels and HAT status at enrolment.**

| A | | Adjusted B | Std Error | P-value |
|---|---|---|---|---|
| | sHLA-G level (log + 1) | | | |
| | CTR Vs HAT | 0.90 | 0.19 | $3.40.10^{-6}$ |
| | SERO Vs HAT | 0.41 | 0.17 | 0.016 |
| | CTR Vs SERO | 0.49 | 0.16 | $3.78.10^{-3}$ |
| B | | Adjusted B | Std Error | P-value |
| | sHLA-G level (log + 1) | | | |
| | CTR vs S1 | 0.93 | 0.28 | $1.37.10^{-3}$ |
| | SERO/TL- vs S1 | 0.58 | 0.29 | 0.047 |
| | CTR Vs S2 | 0.93 | 0.19 | $3.83.10^{-6}$ |
| | SERO/TL- Vs S2 | 0.57 | 0.18 | $1.95.10^{-3}$ |
| | SERO/TL- Vs SERO/TL+ | 0.87 | 0.29 | $2.94.10^{-3}$ |
| | CTR Vs SERO/TL+ | 1.22 | 0.29 | $4.89.10^{-5}$ |
| | CTR Vs SERO/TL- | 0.35 | 0.17 | 0.037 |

A) in general group. B) in specific subgroups. CTR, control; HAT, human African trypanosomiasis cases; SERO, seropositive; TL, trypanolysis; S, stage; sHLA-G, soluble human leucocyte antigen-G. Linear regression was applied to investigate the association between disease status (CTR, SERO, SERO/TL-, SERO/TL +, HAT, S1 and S2) and sHLA-G plasmatic level adjusted on HAT focus, age, sex, fever, pruritus and dermatitis status. Significant results at P <0.05.

plasmatic sHLA-G concentrations were significantly associated with the presence (0 Vs. 1) of HLA-G diffuse halos in the dermis (p = 0.03) (Table 6A). Moreover, individuals with high levels of HLA-G diffuse halos in the dermis (categorical) were significantly associated with high levels of sHLA-G in plasma (p = 0.04) (Table 6B). For the follow-up visits, linear regression mixed models showed no significant association between plasmatic sHLA-G concentrations, and the proportions or delta means of neither HLA-G granular brown spots nor HLA-G diffuse halos in the dermis.

**Table 3. Plasmatic and dermal HLA-G levels in all samples at all visits.**

| Phenotypic groups | | Plasmatic sHLA-G | Dermal HLA-G diffuse halos in IHC | Dermal HLA-G granular brown spots in IHC | Presence of dermal trypanosomes in IHC |
|---|---|---|---|---|---|
| | | Number | Number | Number | Number |
| | | Mean [SEM], ng/ml | % (No.) | % (No.) | % (No.) |
| **CTR** | **All** | n=81 | n=65 | n=65 | n=57 |
| | | 8.25 [9.17] | Neg: 33.85 (22) | Neg: 53.85 (35) | Neg: 71.93 (41) |
| | | | Pos: 66.15 (43) | Pos: 46.15 (30) | Pos: 28.07 (16) |
| **SERO** | **All** [a] | n=82 | n=79 | n=79 | n=62 |
| | | 13.14 [12.53] | Neg: 49.37 (39) | Neg: 55.7 (44) | Neg: 66.13 (41) |
| | | | Pos: 50.63 (40) | Pos: 44.3 (35) | Pos: 33.87 (21) |
| | **SERO TL-** | n=51 | n=48 | n=48 | n=40 |
| | | 8.95 [5.82] | Neg: 50 (24) | Neg: 52.08 (25) | Neg: 62.50 (25) |
| | | | Pos: 50 (24) | Pos: 47.92 (23) | Pos: 37.50 (15) |
| | **SERO TL+** | n=24 | n=24 | n=24 | n=19 |
| | | 14.08 [11.56] | Neg: 45.83 (11) | Neg: 54.17 (13) | Neg: 68.42 (13) |
| | | | Pos: 54.17 (13) | Pos: 45.83 (11) | Pos: 31.58 (6) |
| **HAT** | **All** | n=38 | n=34 | n=34 | n=27 |
| | | 17.73 [12.58] | Neg: 52.94 (18) | Neg: 47.06 (16) | Neg: 81.48 (22) |
| | | | Pos: 47.06 (16) | Pos: 52.94 (18) | Pos: 18.52 (5) |
| | **S1** | n=8 | n=8 | n=8 | n=5 |
| | | 19.44 [16.55] | Neg: 62.5 (5) | Neg: 75 (6) | Neg: 80.00 (4) |
| | | | Pos: 37.5 (3) | Pos: 25 (2) | Pos: 20.00 (1) |
| | **S2** | n=30 | n=26 | n=26 | n=22 |
| | | 17.27 [11.6] | Neg: 50 (13) | Neg: 38.46 (10) | Neg: 81.82 (18) |
| | | | Pos: 50 (13) | Pos: 61.54 (16) | Pos: 18.18 (4) |
| **Total** | | n=201 | n=178 | n=178 | n=146 |
| | | 12.04 [11.79] | Neg: 44.38 (79) | Neg: 53.37 (95) | Neg: 71.23 (104) |
| | | | Pos: 55.62 (99) | Pos: 46.63 (83) | Pos: 28.77 (42) |

Results obtained from all samples collected in all groups at all three timepoints (enrollment, follow-up 1 and follow-up 2) were included in this analysis. CTR, control; SERO, seropositive; TL, trypanolysis; HAT, human African trypanosomiasis; S1, stage 1; S2, stage 2; SEM, Standard error of the mean; sHLA-G, soluble human leukocyte antigen-G; Neg, negative; Pos, positive. [a] Seven diagnostic points in the SERO group had an undetermined TL status.

## Discussion

### Plasmatic levels of sHLA-G and HAT

Here, we investigated the HLA-G expression levels in the blood and skin of seronegative controls, unconfirmed seropositive subjects and confirmed HAT cases from Guinea and Côte d'Ivoire. The implication of sHLA-G plasma level as a biomarker for *T. b. gambiense* infection was confirmed. The results showed significantly higher titers of plasma

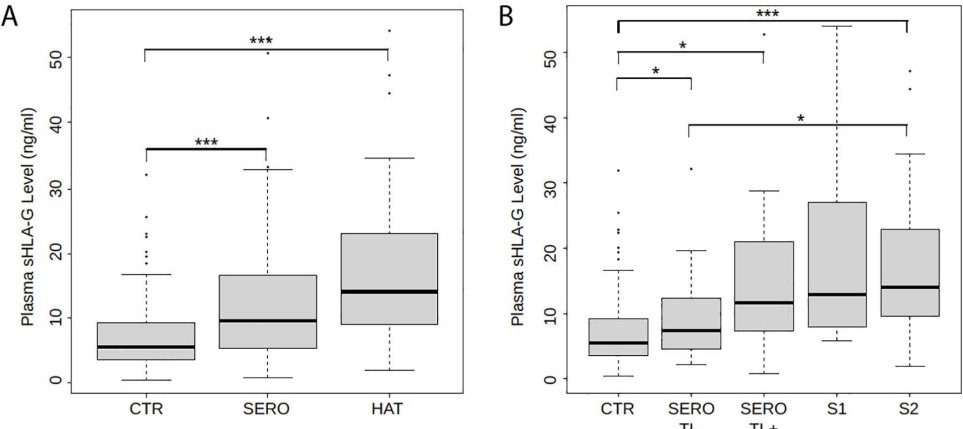

**Fig 3. Plasmatic sHLA-G concentrations by HAT status during the follow-up visits. A)** Plasma levels in control (CTR), seropositive (SERO) and HAT groups. **B)** Plasma levels in CTR, SERO and negative trypanolysis test (SERO/TL-), SERO and positive trypanolysis test (SERO/TL+), stage 1 (S1) and stage 2 (S2) subgroups. *P < .05, ** P < .01, *** P < .001 (P values based on multinomial linear regression for associations between explanatory variables sHLA-G level and HAT status adjusted on sex, age, focus, fever, pruritus and dermatitis status).

**Table 4. Associations between sHLA-G plasmatic levels and HAT status at all visits.**

| A | | Adjusted B | Std Error | P-value |
|---|---|---|---|---|
| | sHLA-G level (log + 1) | | | |
| | **CTR Vs HAT** | **0.90** | 0.64 | **1.80.10⁻⁴** |
| | SERO Vs HAT | 0.18 | 0.15 | 0.22 |
| | **CTR Vs SERO** | **0.45** | 0.13 | **8.13.10⁻⁴** |
| B | | Adjusted B | Std Error | P-value |
| | sHLA-G level (log + 1) | | | |
| | **CTR Vs S1** | **0.58** | 0.30 | **0.053** |
| | SERO/TL- Vs S1 | 0.26 | 0.30 | 0.39 |
| | **CTR Vs S2** | **0.76** | 0.18 | **3.37.10⁻⁵** |
| | **SERO/TL- Vs S2** | **0.45** | 0.19 | **0.02** |
| | SERO/TL- Vs SERO/TL+ | 0.18 | 0.20 | 0.36 |
| | **CTR Vs SERO/TL+** | **0.49** | 0.19 | **0.01** |
| | **CTR Vs SERO/TL-** | **0.32** | 0.15 | **0.033** |

A) in general group. B) in specific subgroups. CTR, control; HAT, human African trypanosomiasis; SERO, seropositive; TL, trypanolysis; S, Stage; sHLA-G, soluble human leucocyte antigen-G. Linear mixed regression was applied to investigate the association between explanatory variables HAT status (CTR, SERO/TL-, SERO/TL +, S1 and S2) and sHLA-G plasmatic levels adjusted on HAT focus, age, sex, fever, pruritus and dermatitis status. Significant results at P < 0.05.

sHLA-G in confirmed HAT cases as compared to the CTR group. Moreover, a higher sHLA-G level was observed in the subgroup SERO/TL+ than the CTR group, suggesting an increased risk of developing the HAT for these seropositive suspects. Indeed, high plasma levels of sHLA-G in unconfirmed seropositive subjects for gHAT have been previously correlated with HAT disease progression [30]. The quantification of similar plasma HLA-G levels in the SERO/TL +, S1 and S2 subgroups strongly suggests a risk of latent infection in this group of serological suspects. Due to its strong immunosuppressive properties [29], HLA-G was considered as a good candidate gene and the role played by *HLA-G* genetic polymorphisms in the variable clinical outcomes of gHAT has been studied [29]. A human genetic study

**Table 5. Associations between HLA-G detected in the dermis by mean of delta granular brown spots and parasitological status in the dermis assessed by immuno-histochemistry or dermatitis status.**

| A | | Adjusted B[(a)] | Std Error | P-value |
|---|---|---|---|---|
| | **Delta HLA-G granular brown spots (0 vs 1)** | | | |
| | **IHC (anti-ISG65)** | **1.68** | 0.57 | **$3.16.10^{-3}$** |
| | **Dermatitis status** | **1.81** | 0.62 | **$3.63.10^{-3}$** |
| B | | Adjusted B[(b)] | Std Error | P-value |
| | **Delta HLA-G granular brown spots (0 vs 1)** | | | |
| | **IHC (anti-ISG65)** | **0.92** | 0.40 | **$2.5.10^{-3}$** |
| | **Dermatitis status** | **1.02** | 0,41 | **$1.02.10^{-2}$** |

A) at enrolment point. B) during follow-up. IHC, immuno-histochemistry. [a] Logistic regression and [b] logistic mixed regression were applied to investigate the association between explanatory variables IHC and HLA-G in the dermis detected as granular brown spots adjusted on HAT focus, age, sex, fever, pruritus and dermatitis status. Significant results at P<0.05.

**Table 6. Associations between sHLA-G plasmatic levels and HLA-G detected in the dermis by mean of diffuse halos at enrollment.**

| A | | | Adjusted B[(a)] | Std Error | P-value |
|---|---|---|---|---|---|
| | **sHLA-G level (log+1)** | | | | |
| | **Delta HLA-G diffuse halos (0 vs 1)** | | | | |
| | | | **0.31** | 0.14 | **0.03** |
| | | | Adjusted B[(a)] | Std Error | P-value |
| B | **sHLA-G level (log+1)** | | | | |
| | **Delta HLA-G diffuse halos (categorical)** | | | | |
| | | Null vs Medium level | 0.26 | 0.16 | 0.11 |
| | | **Null vs High level** | **0.35** | 0.17 | **0.04** |

A) with dichotomic variable. B) categorial variable. sHLA-G, soluble human leucocyte antigen-G. [a] Linear regression was applied to investigate the association between explanatory variables sHLA-G plasmatic level and HLA-G in the dermis detected as diffuse halos adjusted on HAT focus, age, sex, fever, pruritus and dermatitis status. Significant results at P<0.05.

performed in Democratic Republic of Congo revealed an association between the presence of *HLA-G* UTR-2 haplotype and an increased risk of developing HAT [40]. This result was the first to report an association between *HLA-G* polymorphisms and the risk of developing HAT and suggested the involvement of the HLA-G molecule on HAT susceptibility. Indeed, polymorphisms in HLA-G 3′UTR may potentially influence HLA-G transcription, translation, or both, by several mechanisms such as additional alternative splicing or miRNAs fixation [41]. Associations between *HLA-G* and HAT susceptibility were replicated in Côte d'Ivoire [18], Cameroon [16] and Uganda [42] by the TRYPANOGEN research group from the H3Africa consortium. It should be noted that in the Ugandan study, the genetic association was highlighted between *HLA-G* and both forms of HAT (caused by *T.b. gambiense* and *T.b. rhodesiense*). Interestingly, other *HLA-G haplotypes* (HG010102/HG0103) were associated with resistance or susceptibility to disease development at the neurological stage [30]. Our results showed no significant difference of plasmatic sHLA-G concentrations between subjects in stage 1 and in stage 2 of the disease. High CSF levels of sHLA-G were previously associated with a higher probability of developing late stage 2 disease [30], unfortunately the quantification of HLA-G in CSF was not performed in the present study.

## Dermal levels of HLA-G and HAT

As discussed in [14], the limited sensitivity of IHC on superficial skin snip biopsies appeared critical for detecting trypano-somes in the basal dermis of confirmed cases and seropositive suspects. This could likely explain the lowest positivity rate observed here and in [14], as compared to our previous study using deep skin punch biopsies [13]. Regarding the samples positive for dermal trypanosomes in the control group, it is noteworthy that some individuals classified as sero-negative controls in the present study, according to their test results at enrollment (negative in CATTwb and/ or RDT, and negative in CATTp), were part of the cohort of gHAT suspects followed up in Côte d'Ivoire for their atypical serological profiles over years [15]. Hence, the present study focuses on investigating the differences in HLA-G expression in the skin of these subjects as compared to their plasmatic sHLA-G levels.

The HLA-G quantification in the dermal tissue was tedious, due to the limited signal intensity, which resulted in a limited sensitivity. However, the use of internal control slides for normalization of the signal (deltas between two consecutive unstained and labeled slides) combined to the use of a highly specific 4H84 antibody guaranteed a good specificity of our approach. In the dermis of the present subjects, HLA-G isoforms were expressed either with a granular distribution, pos-sibly reflecting their localization at the surface of immune cells, or with in diffuse halos, probably reflecting accumulation of soluble HLA-G molecules in the interstitial fluid of the dermis. In our study, higher sHLA-G plasmatic concentrations were significantly associated with the presence of HLA-G diffuse halos in the dermis, and individuals with high levels of HLA-G diffuse halos in the dermis were significantly associated with high levels of sHLA-G in plasma. This tends to confirm the soluble nature of the diffuse HLA-G signal, with a continuum in the composition of the plasma and interstitial fluids in the dermis.

While the presence of detectable parasites in the dermis was not directly correlated to the amount of HLA-G diffuse halos in the dermis, granular patterns of dermal HLA-G were directly associated with the detection of trypanosomes in the dermis. This suggests a role of transmembrane HLA-G isoforms in an inflammatory immune response against the parasites in the dermis. The human cells present in the dermis are mainly fibroblasts, sensory cells and immune cells. Under physiological conditions, fibroblasts and sensory cells are not known to express HLA-G molecules and only a small proportion of HLA-G-positive immune cells can be detected in the peripheral blood. However, the proportions of circu-lating and tissue infiltrating HLA-G-positive immune cells frequently increase in various pathological conditions such as immune-mediated diseases or infections [43]. In allergic patients with atopic dermatitis, HLA-G molecules were expressed by T cells, Langerhans cells, and monocytes-macrophages infiltrating cells in the dermis [44]. *In vitro* experiment per-formed on peripheral blood mononuclear cells from allergic rhinitis patients showed an increase HLA-G expression by CD4+T lymphocytes and monocytes during 72 h of incubation with the causal allergen [45]. Works carried on HLA-G expression by immune cells during infections focused mainly on the blood compartment. Elevated sHLA-G plasma levels were detected in patients with chronic HCV infection [46], HBV infection [47,48] when compared to healthy individuals. An increase in the percentage of monocytes and T cells expressing HLA-G was observed in the presence of HIV-1 than in healthy individuals [49,50]. In total, as suggested for viral infections, the HLA-G up-regulation observed at the surface of immune cells in presence of trypanosome infection may be a way to modulate the immune surveillance level, or part of an internal regulatory system to control excessive inflammation.

This hypothesis is reinforced by the significant association observed between the positivity for HLA-G granular brown spots in a dermal sample and the occurrence of dermatitis in the subject. Indeed, the few reports that exist on this topic in the literature describe a wide array of skin pathologies associated with HAT, including chancre, rashes, localized edemas, and more frequently pruritus [51,52]. Recently, we confirmed that dermatological symptoms such as pruritus and derma-titis were significantly more frequent in seropositive suspects and confirmed cases bearing dermal trypanosomes [13]. At that time, the observed dermatitis profiles included some conditions the etiologies of which were not directly imput-able to a trypanosome infection. However, we hypothesized that the immune status of the infected host skin could have somehow been altered by the presence of trypanosomes in a way that would have promoted the outcome of dermatitis

caused by other pathogens and/or increases skin sensitivity. In the light of the present study, we could now propose that an increased number of monocytes and T cells expressing HLA-G in response to a trypanosome infection in the dermis could be involved in this complex and dynamic modulation. Unfortunately, a co-detection of both HLA-G molecules and parasites markers at the same time was technically not possible. The methodology used to detect HLA-G expression in the present study was also not able to identify the HLA-G-expressing cells in the dermis and the different HLA-G isoforms. Therefore, the contribution of HLA-G expressing cells and the role played by different HLA-G isoforms during trypanosome infections needs to be better defined in the future.

Overall, this study provides the first evidence of the involvement of HLA-G in the extravascular immune response against parasites. It shows that HLA-G distribution in the extravascular compartment also represents a biomarker of trypanosome infection. Understanding the interplay between HLA-G, dermal immunity, and plasma levels in the context of HAT could provide valuable insights into disease mechanisms and inform the development of novel therapeutic strategies.

## Supporting information

**S1 Fig. Immuno-histochemical detection of HLA-G in human placenta.** As positive controls, formalin-fixed paraffin-embedded human placenta sections were immunolabelled with the anti-HLA-G-4H84 antibody targeting both soluble and membrane HLA-G molecules. (A) Image of an entire section showing a strong labelling in the extravillous cytotrophoblasts at the periphery of the placenta. (B) Higher magnification images of extravillous cytotrophoblasts stained for HLA-G. Scale bars represents 20 μm.
(EPS)

**S2 Fig. Flow-charts of the population study.** A) Number of subjects per time-point per clinical site. B) Number of subjects per time-point per study group.
(EPS)

**S1 Table. Diagnostic tests and group definition.** CATTwb/ CATTp: card agglutination test for trypanosomiasis on whole blood/ plasma; TL: immune trypanolysis test; RDT: Rapid Diagnostic Test for HAT; mAECT BC/ LN aspirate: mini anion-exchange column technique on buffy coat/ lymph node aspirate; WBC: white blood cells; CSF: cerebrospinal fluid; *Highest plasma dilution with a positive result; S1 and S2: HAT stage 1 and 2.
(DOCX)

**S2 Table. Associations between parasitological status in the blood and in the dermis.** A) at enrolment and B) during follow-up, and between result of trypanolysis test and parasitological status in the dermis C) at enrolment and D) during follow-up. [a] Logistic regression and [b] logistic mixed regression were applied to investigate the association between explanatory variables (IHC, HAT focus, age, sex, fever, pruritus and dermatitis) and parasitology (A and B) or trypanolysis (C and D) status in the blood. Significant results at P<0.05. mAECT BC/ LN aspirate: mini anion-exchange column technique on buffy coat/ lymph node aspirate; IHC: immuno-histochemistry.
(DOCX)

**S3 Table. Associations between sHLA-G plasmatic level and parasitological status in the blood.** A) at enrolment. B) during follow-up. sHLA-G, soluble human leucocyte antigen-G. [a] Linear regression and [b] linear mixed regression were applied to investigate the association between explanatory variables parasitological status in the blood (mAECT) and sHLA-G plasmatic level adjusted on HAT focus, age, sex, fever, pruritus and dermatitis status. Significant results at P<0.05.
(DOCX)

**S4 Table. Associations between sHLA-G plasmatic level and results of trypanolysis tests.** A) at enrolment. B) during follow-up. sHLA-G, soluble human leucocyte antigen-G. [a] Linear regression and [b] linear mixed regression were

applied to investigate the association between explanatory variables trypanolysis test result and dermatitis adjusted on HAT focus, age, sex, fever, pruritus and dermatitis status. Significant results at P<0.05.
(DOCX)

**S1 File. Source data.** The data file presents all the source data used in this study. Group 1: main groups; Group 2: with sub-groups based on immune trypanolysis test results (TL) and HAT stages; Timepoint: enrolment, follow-up 1 or follow-up 2; Screening code: from the national programmes; TrypaDerm ID: individual identifier for the present study; Focus: Bonon, Boffa, Forecariah or Sinfra; Delta HLA-G halos: Delta number of HLA-G halos as compared to the paired control slide; Delta HLA-G granular spots: Delta number of HLA-G granular spots as compared to the paired control slide; Mean sHLA-G concentration: in ng/ ml; Mean.concentration_CI: 95% confidence intervalle in ng/ ml; Age: in years; Sex: female (0) or male (1); HAT.cases.in.the.family: occurrence of any confirmed HAT cases in the family since 2010; Swollen.LN: occurrence of swollen cervical lymph nodes; CATTwb: result of the card agglutination test for trypanosomiasis on whole blood; CATTp: result of the card agglutination test for trypanosomiasis on plasma with an end titer of at least 1/4; TDR.SD.HAT.result: result of the associated rapid diagnostic test for gHAT on blood (SD Bioline HAT, Abbott Bioline HAT 2.0 or HAT Sero-K-SeT); LN.aspirate.result: result of the parasitological observation of the lymph node aspirate; mAECTbc.result: result of the mini-anion exchange centrifugation test; mAECTbc.number.parasite: number of parasites detected in mini-anion exchange centrifugation test; CSF.result: result of the parasitological observation of the cerebrospinal fluid; CSF.result.number.parasite: number of parasites detected in parasitological observation of the cerebrospinal fluid; No.WBC.in.CSF: number of white blood cells detected in the cerebrospinal fluid; LiTat.1.3, LiTat.1.5, LiTat.1.6, and TL.total: results of the immune trypanolysis tests (TL) for detecting each or all variable antigen types, respectively; ISG65. IHC.FINAL.High.thresholds: results of the immunohistochemistry analysis with the anti-ISG65 detecting *T. brucei* parasites in skin sections; 1: yes/ presence/ positive; 0: no/ absence/ negative; NA: not available/ not done; NI: not interpretable.
(XLSX)

## Acknowledgments

We warmly thank the team of the Programme National de Lutte contre la Trypanosomiase Humaine Africaine of Guinea, as well as all our collaborators of the Forecariah and Boffa Health Districts. We warmly thank the team of the Programme National d'Elimination de la Trypanosomiase Humaine Africaine of Côte d'Ivoire, as well as all our collaborators of the Sinfra and Bouaflé Health Districts. We thank M. Carrington (University of Cambridge, UK) for providing the anti-ISG65 antibody, and Nadine Fievet who provided the human placenta sections used as positive control.

## Author contributions

**Conceptualization:** David Courtin, Brice Rotureau.

**Data curation:** Alisé Lagrave, Aïssata Camara, Laure Gineau, David Courtin.

**Formal analysis:** Alisé Lagrave, Aïssata Camara, Laure Gineau, David Courtin, Brice Rotureau.

**Funding acquisition:** Bruno Bucheton, David Courtin, Brice Rotureau.

**Investigation:** Alisé Lagrave, Aïssata Camara, Laure Gineau, Magali Tichit, Firmin Bolivar Gnankou, Alseny M'mah Soumah, Mariame Camara, Martial N'Djetchi, Justin Windingoudi Kaboré, Oumou Camara, Bamoro Coulibaly, Blé Sépé, Valentin Nanan, Koffi Alain De Marie Kouadio, Louis N'Dri, Thomas Konan, Jacqueline Milet, Salimatou Boiro, Christelle Travaillé, Aline Crouzols, Nathalie Petiot, Hamidou Ilboudo, David Hardy, Ibrahim Sadissou, Jean-Mathieu Bart, Bruno Bucheton, Vincent Jamonneau, David Courtin, Brice Rotureau.

**Methodology:** Alisé Lagrave, Laure Gineau, David Hardy, David Courtin, Brice Rotureau.

**Project administration:** David Hardy, Mamadou Camara, Dramane Kaba, Mathurin Koffi, David Courtin, Brice Rotureau.

**Resources:** David Hardy, Mamadou Camara, Dramane Kaba, Mathurin Koffi, Vincent Jamonneau, David Courtin, Brice Rotureau.

**Supervision:** David Hardy, Jean-Mathieu Bart, Mamadou Camara, Dramane Kaba, Mathurin Koffi, Bruno Bucheton, Vincent Jamonneau, David Courtin, Brice Rotureau.

**Validation:** Aïssata Camara, Laure Gineau, David Courtin, Brice Rotureau.

**Visualization:** Laure Gineau.

**Writing – original draft:** Alisé Lagrave, Aïssata Camara, Laure Gineau, David Courtin, Brice Rotureau.

**Writing – review & editing:** Aïssata Camara, Laure Gineau, Bruno Bucheton, David Courtin, Brice Rotureau.

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
