## [Decision Letter · Decision Letter 0]

26 Oct 2025

Human Leukocyte Antigen-G is enriched in presence of trypanosome in the dermis of individuals exposed to gambiense human African trypanosomiasis in Guinea and Côte d’Ivoire.

Dear Dr. Rotureau,

Thank you for submitting your manuscript to PLOS Neglected Tropical Diseases. After careful consideration, we feel that it has merit but does not fully meet PLOS Neglected Tropical Diseases's publication criteria as it currently stands. Therefore, we invite you to submit a revised version of the manuscript that addresses the points raised during the review process.

Please submit your revised manuscript within 60 days Dec 25 2025 11:59PM. If you will need more time than this to complete your revisions, please reply to this message or contact the journal office at plosntds@plos.org. Please include the following items when submitting your revised manuscript:

We look forward to receiving your revised manuscript.

Kind regards,

Jayne Raper, PhD

Academic Editor

Guilherme Werneck

Section Editor

Shaden Kamhawi

co-Editor-in-Chief

Paul Brindley

co-Editor-in-Chief

**Additional Editor Comments :**

Brice

Thank you for your submission to PLoS NTD, super interesting finding. I turned to three reviewers with different expertise for this paper, which is reflected in their comments. I agree with them. Some of their comments overlap, indicating an issue. Please address these issues.

I know image acquisition is not easy, but consider image processing. label your images appropriately.

There is clarity requested in your methodology please address these issues.

Include all raw data as requested in supplemental data.

I look forward to seeing your revised manuscript.

Jayne

**Journal Requirements:**

**Comments to the Authors:**

**Please note that one review is uploaded as an attachment.**

**Reviewers' Comments:**

Reviewer's Responses to Questions

**Key Review Criteria Required for Acceptance?**

**Methods**

-Are the objectives of the study clearly articulated with a clear testable hypothesis stated?

-Is the study design appropriate to address the stated objectives?

-Is the population clearly described and appropriate for the hypothesis being tested?

-Is the sample size sufficient to ensure adequate power to address the hypothesis being tested?

-Were correct statistical analysis used to support conclusions?

-Are there concerns about ethical or regulatory requirements being met?

Reviewer #1: The methods are very clearly described - the authors have done an very good job of describing the technical details of their wet-lab experiments as well as their patient samples. The N numbers in each group are a key strength of the manuscript, given that virtually 100% of the data and interpretations are based on statistics. The regression models chosen for each analysis are good choices as far as I can tell.

Reviewer #2: Please see the attached file with the recommendations.

Reviewer #3: The objectives of the study are clearly articulated, with a clear testable hypothesis.

The study design could improve, particularly the figures including images, as they are being included in support of the main hypothesis.

The population is clearly described, and is appropriate for the hypothesis being tested.

The sample size is sufficient, and the statistical analysis is valid. Nevertheless, there are important elements of figures 1 and S1 missing - and the information included in the layout should improve too.

I did not find anything concerning regarding ethical or regulatory elements.

**Results**

-Does the analysis presented match the analysis plan?

-Are the results clearly and completely presented?

-Are the figures (Tables, Images) of sufficient quality for clarity?

Reviewer #1: The manuscript would benefit from higher quality images as well as a higher number of images. This is a significant issue with the manuscript. The image quality in figure 1 is poor to the point where even the computer-drawn arrows look pixelated. Presumably this was a file size issue? But in any case, the reader is left wondering how many of these IHC signals are actually real, which is particularly problematic in the context of the "diffuse halos" which are really weak signals (to the authors' credit - they point this out themselves and address it in the discussion). My suggestion would be to 1) improve the figure quality / sharpness and 2) add in several more positive staining samples to the supplemental figures so that the reader can see for themselves, rather than rely only on the numbers in the tables.

And speaking to the tables - it would be much much nicer to present the key linear regressions as actual charts for the reader to evaluate. Only publishing the B and P values for all of these data is less satisfying for the reader.

Reviewer #2: Please see the attached file with the recommendations.

Reviewer #3: The analysis presented in the tables is thorough, and matches the analysis plan.

Regarding the results, I have concerns with current Figure 1 and Figure S1. The authors address the issue of poor signal. I would suggest doing image processing - for instance, CLAHE (a plugin for Fiji) would help. At present, the point they aim to make is not very clear. The design of these figures could also improve as follows:

a) There should be representation of more than one sample and this should be clearly shown in the figure.

b) Currently, it is unclear whether the larger magnification/zoom-in come from the low magnification image shown, or not, if they are separate (which ideally, they should not be). The area of the zoom in should be clearly shown.

c) Scale bars are missing. They must be included in order to help the reader better understand the images.

d) Improving the image quality through pre-processing might allow a more quantitative approach.

e) Although the authors mention that the study is semi-quantitative regarding the images, and although the tables show these quantifications, it would be useful to include even the semi-quantitative graphs together with the figure - it helps the reader evaluate each figure standalone, as well as in the context of the tables and the general paper.

f) Where/if arrows are used, for the inserts, the arrows in the low magnification figure should be numbered. That number would correspond to the high-mag/zoom-in image. Right now it lacks structure.

g) The authors mention that it was not possible to have co-detection of HLA-G and parasite markers. It is not clear in the text why this is the case.

h) If a fully quantitative approach is possible for the images, clear methodology should be included as to how the analysis was performed.

Another key point is that the raw data (for both ELISAs and imaging) that were used to derive the conclusions of the paper, should be included.

**Conclusions**

-Are the conclusions supported by the data presented?

-Are the limitations of analysis clearly described?

-Do the authors discuss how these data can be helpful to advance our understanding of the topic under study?

-Is public health relevance addressed?

Reviewer #1: There are three main conclusions in this paper.

1) the authors observe supportive data for a pre-existing conclusion that serum-HLAg levels coorelate with trypanosome infection. The correlations presented in this manuscript in this context are strong and indeed it appears that serum-HLAg could be considered a good biomarker in the Tbg infection context (though not a Tbg-specific biomarker - as the authors admit). I would prefer that the data be shown as individual data points rather than box-whiskers though. Easier for the reader to determine how strong it really is. Box-whisker for me is only appropriate when you have really very high N numbers.

2) the authors observe that the "granular brown spots" of HLAg in the dermis correlate with the presence of trypanosomes. The correlation here is again strong. The expectation is that these HLAg signals are from cell-bound / transmembrane HLAg on some skin-infiltrating immune cells, which is also observed in other skin conditions.

The authors state that they were unable to investigate the cells expressing this HLAg. IHC does have its limitations and I do not think additional experiments are necessary for publication here; but I would say that the novelty of this finding would be significantly more impactful if the identity of these cells was determined. Presumably the extensive trypanosome literature could be consulted, identifying a small handful of candidate cell types that could be stained for in follow up studies. But indeed, these would need to be immunofluorescence studies instead of IHC (for colocalization). If the authors are capable / equipped for such studies, then the manucsript would be much improved.

Would the authors hypothesize that the HLAg expressing cells are playing some immunomodulatory role that helps the trypanosome infection remain "dormant" in the skin? Or would they hypothesize that this HLAg is directly trypanosome-suppressive? The authors touch on these topics in the discussion (lines 445-466) but I'm not sure what their hypotheses would be.

3) the authors observe that the "diffuse halos" of HLAg in the dermis correlate with the serum HLAg levels. This correlation is much much weaker than those above as far as I can tell from the P values. And given that the signal intensities from the "diffuse halos" are so weak, I'm hesitant to trust this correlation. From a logical perspective, a higher amount of circulating HLAg could very well facilitate the "leakage" of HLAg into the dermis so long as there is some mechanism for its transport out of the blood vessels. Is that where it is "coming from"? Or is the hypothesis that it is locally produced and secreted within the dermis? ... Nevertheless, I think the data supporting conclusion 3 are weak, but not so weak to prevent publication. I do however think that publishing a lot more of the raw data (many more of the IHC images) in the supplementals so that the reader can make their own interpretations is key.

Reviewer #2: Please see the attached file with the recommendations.

Reviewer #3: The conclusions are supported by the data, but better images will help make the point. The limitations of the analysis are well described. The public health relevance is addressed, and the authors discuss how the data can help us advance our understanding of the topic. As mentioned above though, the figures including images are not 100% convincing - by both B&C and a relative lack of clarity in the annotations shown.

**Editorial and Data Presentation Modifications?**

Reviewer #1: Improved figure quality as discussed

Typos throughout that must be fixed.

Reviewer #2: (No Response)

Reviewer #3: As suggested above - Figures 1 and S1 can be improved in quality, annotation and content.

The raw data of both ELISAs and images, should be included.

**Summary and General Comments**

Reviewer #1: Working with patient-sourced tissues carries with it a unique set of challenges. I think the authors can do a bit better with the way they present some of the data (more IHC images and better image quality) but overall I think they have put together a nice manuscript.

Reviewer #2: The study provides promising data regarding the role of HLA-G in HAT. However, essential revisions are needed to improve methodological clarity, data presentation, and control group selection. Once these issues are addressed, the manuscript will be better positioned to make a significant contribution to the field.

Reviewer #3: Overall, the paper by Lagrave et al is of great relevance, and of significant novelty in the context of better understanding of human African trypanosomiasis. They use a combination of immuno-histochemistry and ELISA to derive the conclusions shown. In general, the results are interesting. Major revision is requested, particularly in relation to Figures 1 and S1 (see above comments).

PLOS authors have the option to publish the peer review history of their article (what does this mean? ). If published, this will include your full peer review and any attached files.

**Do you want your identity to be public for this peer review?** For information about this choice, including consent withdrawal, please see our Privacy Policy .

Reviewer #1: No

Reviewer #2: No

Reviewer #3: No

**Figure resubmission:**
---

## [Decision Letter · Decision Letter 1]

24 Feb 2026

Dear Dr. Rotureau,

We are pleased to inform you that your manuscript 'Human Leukocyte Antigen-G is enriched in presence of trypanosome in the dermis of individuals exposed to gambiense human African trypanosomiasis in Guinea and Côte d’Ivoire.' has been provisionally accepted for publication in PLOS Neglected Tropical Diseases.

Best regards,

Guilherme L Werneck

Section Editor

Guilherme Werneck

Section Editor

Shaden Kamhawi

co-Editor-in-Chief

Paul Brindley

co-Editor-in-Chief

Reviewer's Responses to Questions

**Key Review Criteria Required for Acceptance?**

**Methods**

-Are the objectives of the study clearly articulated with a clear testable hypothesis stated?

-Is the study design appropriate to address the stated objectives?

-Is the population clearly described and appropriate for the hypothesis being tested?

-Is the sample size sufficient to ensure adequate power to address the hypothesis being tested?

-Were correct statistical analysis used to support conclusions?

-Are there concerns about ethical or regulatory requirements being met?

Reviewer #1: Methods are satisfactory.

Reviewer #2: Accept

Reviewer #3: The objectives of the study are clearly articulated, with clear and testable hypotheses. The study design is appropriate to address the stated objectives. The authors improved the manuscript significantly and addressed the recommendations of the reviewers.

**Results**

-Does the analysis presented match the analysis plan?

-Are the results clearly and completely presented?

-Are the figures (Tables, Images) of sufficient quality for clarity?

Reviewer #1: Results section has been improved by virtue of the figure quality and inclusion of the raw data in file S1.

Reviewer #2: Accept

Reviewer #3: The results section is correct. Additional tables and raw data were included, as requested by the reviewers.

**Conclusions**

-Are the conclusions supported by the data presented?

-Are the limitations of analysis clearly described?

-Do the authors discuss how these data can be helpful to advance our understanding of the topic under study?

-Is public health relevance addressed?

Reviewer #1: The conclusions are supported well enough.

Reviewer #2: Accept

Reviewer #3: The conclusions support the data presented. From my area of research, I find the manuscript ready, and of high relevance to the community.

**Editorial and Data Presentation Modifications?**

Reviewer #1: N/A

Reviewer #2: Accept

Reviewer #3: (No Response)

**Summary and General Comments**

Reviewer #1: The manuscript is improved and ready for publication.

Reviewer #2: Accept

Reviewer #3: The authors greatly improved the manuscript and addressed all the reviewers' points. The paper is of great relevance to the scientific community, and is ready for acceptance.

PLOS authors have the option to publish the peer review history of their article (what does this mean? ). If published, this will include your full peer review and any attached files.

**Do you want your identity to be public for this peer review?** For information about this choice, including consent withdrawal, please see our Privacy Policy .

Reviewer #1: No

Reviewer #2: No

Reviewer #3: No

---

## [Editor Report · Acceptance letter]

Dear Dr. Rotureau,

We are delighted to inform you that your manuscript, "Human Leukocyte Antigen-G is enriched in presence of trypanosome in the dermis of individuals exposed to gambiense human African trypanosomiasis in Guinea and Côte d’Ivoire.," has been formally accepted for publication in PLOS Neglected Tropical Diseases.

Best regards,

Shaden Kamhawi

co-Editor-in-Chief

Paul Brindley

co-Editor-in-Chief
